# Comparative molecular genomic analyses of a spontaneous rhesus macaque model of mismatch repair-deficient colorectal cancer

Nejla Ozirmak Lermi[1,2], Stanton B. Gray[3], Charles M. Bowen[1], Laura Reyes-Uribe[1], Beth K. Dray[4], Nan Deng[1], R. Alan Harris[5], Muthuswamy Raveendran[5], Fernando Benavides[6], Carolyn L. Hodo[3], Melissa W. Taggart[7], Karen Colbert Maresso[1], Krishna M. Sinha[1], Jeffrey Rogers[5], Eduardo Vilar[1,8]*

1 Department of Clinical Cancer Prevention, The University of Texas MD Anderson Cancer Center, Houston, Texas, United States of America, 2 School of Health Professions, The University of Texas MD Anderson Cancer Center, Houston, Texas, United States of America, 3 Michale E. Keeling Center for Comparative Medicine and Research, The University of Texas MD Anderson Cancer Center, Houston, Texas, United States of America, 4 Charles River Laboratories, Ashland, Ohio, United States of America, 5 Human Genome Sequencing Center and Department of Molecular and Human Genetics, Baylor College of Medicine, Houston, Texas, United States of America, 6 Department of Epigenetics and Molecular Carcinogenesis, The University of Texas MD Anderson Cancer Center, Houston, Texas, United States of America, 7 Department of Pathology, The University of Texas MD Anderson Cancer Center, Houston, Texas, United States of America, 8 Clinical Cancer Genetics Program, The University of Texas MD Anderson Cancer Center, Houston, Texas, United States of America

* EVilar@mdanderson.org

**Data Availability Statement:** The project data has been deposited in GEO. The data sets generated

## Abstract

Colorectal cancer (CRC) remains the third most common cancer in the US with 15% of cases displaying Microsatellite Instability (MSI) secondary to Lynch Syndrome (LS) or somatic hypermethylation of the *MLH1* promoter. A cohort of rhesus macaques from our institution developed spontaneous mismatch repair deficient (MMRd) CRC with a notable fraction harboring a pathogenic germline mutation in *MLH1* (c.1029C<G, p.Tyr343Ter). Our study aimed to provide a detailed molecular characterization of rhesus CRC for cross-comparison with human MMRd CRC. We performed PCR-based MSI testing (n = 41), transcriptomics analysis (n = 35), reduced-representation bisulfite sequencing (RRBS) (n = 28), and *MLH1* DNA methylation (n = 10) using next-generation sequencing (NGS) of rhesus CRC. Systems biology tools were used to perform gene set enrichment analysis (GSEA) for pathway discovery, consensus molecular subtyping (CMS), and somatic mutation profiling. Overall, the majority of rhesus tumors displayed high levels of MSI (MSI-H) and differential gene expression profiles that were consistent with known deregulated pathways in human CRC. DNA methylation analysis exposed differentially methylated patterns among MSI-H, MSI-L (MSI-low)/MSS (MS-stable) and LS tumors with *MLH1* predominantly inactivated among sporadic MSI-H CRCs. The findings from this study support the use of rhesus macaques as an alternative animal model to mice to study carcinogenesis, develop immunotherapies and vaccines, and implement chemoprevention approaches relevant to sporadic MSI-H and LS CRC in humans.

and analyzed during the current study can be accessed for re-analysis using the following link through GEO Series accession number GSE178383. (https://www.ncbi.nlm.nih.gov/geo/query/acc.cgi?acc=GSE178383). To access total RNA-seq data, use the following GEO sub-series accession number GSE178381. (https://www.ncbi.nlm.nih.gov/geo/query/acc.cgi?acc=GSE178381). To access RRBS data, use the following GEO sub-series accession number GSE178377. (https://www.ncbi.nlm.nih.gov/geo/query/acc.cgi?acc=GSE178377).

**Funding:** This work was supported by a gift from the Feinberg Family Foundation and MDACC Institutional Research Grant (IRG) Program to E.V.; MD Anderson Internal Grant Award from Cattlemen for Cancer Research to S.G.; R24 OD011173 (US National Institutes of Health) to J.R.; and CA016672 (US National Institutes of Health/National Cancer Institute) to The University of Texas MD Anderson Cancer Center Core Support Grant. The funders had no role in study design, data collection and analysis, decision to publish, or preparation of the manuscript.

**Competing interests:** I have read the journal's policy and the authors of this manuscript have the following competing interests: Dr. Vilar has a consulting or advisory role with Janssen Research and Development and Recursion Pharma. He has received research support from Janssen Research and Development. Please, note that these financial relations are not connected to the research reported in this manuscript.

## Author summary

CRC remains the third most common cancer diagnosed in the United States. Some CRC may arise spontaneously without any known risk factors, while others may arise in patients with strong family history associated to inherited genetic syndromes. Our study focused on a genetic condition known as Lynch Syndrome (LS), which significantly increases the risk of developing CRC as well as several different types of cancers. Biological tools and laboratory animal models available for studying hereditary CRC remain limited and not always directly translatable to human disease. Therefore, our study presents a comprehensive analysis of a spontaneous non-human primate (NHP) model used to study the genetic contribution in LS CRC. We performed a cross-comparison of different types of CRC in humans with tumors developed in monkeys to determine the accuracy of using this NHP model for studying early cancer development, treatment options, and prevention approaches in both hereditary and sporadic colorectal cancer displaying MMR deficiency.

## Introduction

Colorectal cancer (CRC) remains the third leading cause of cancer-related deaths affecting both men and women [1]. Approximately 15% of CRC cases display microsatellite instability (MSI) secondary to a defective mismatch repair (MMRd) system that is recognized as a major carcinogenic pathway for CRC development. MMRd arises from either (1) an inherited germline mutation in one of four genes (*MLH1*, *MSH2*, *MSH6* and *PMS2*) constituting the DNA MMR system followed by an acquired second hit in the wild-type allele of the same gene in colonic mucosa cells (i.e., Lynch syndrome), or (2) somatic inactivation of the *MLH1* gene (i.e., MSI sporadic CRC).

A better understanding of colorectal neoplasia arising in the setting of MSI/MMRd is urgently needed to tailor the use of early detection, prevention, and treatment interventions in this subset of CRC, including immunotherapies (e.g. checkpoint inhibitors) and the development of novel immuno-preventive regimens (e.g. vaccines). Such interventions are particularly needed for those with Lynch syndrome (LS), as they are at the highest risk of CRC as well as a range of other cancers. Unfortunately, no concrete model with higher translational value exists to study the nuanced carcinogenesis of MMRd CRC, which is a critical barrier for studying this subset of CRC and, consequently, to making advances in its detection, prevention, and treatment.

Presently, *in vitro* and *ex vivo* models, such as cell lines and organoids respectively, are commonly used to study CRC; however, the intrinsic nature of these models lack cellular heterogeneity and fail to recapitulate the tumor microenvironment (TME) observed *in-vivo* [2]. To tackle the limitations of *in-vitro/ex-vivo* cultures, mouse models (*Mus musculus*) have been leveraged to study CRC prevention, initiation, and progression. Although murine models of genetic inactivation of MMR genes exist, these models present critical differences to the human LS (MMRd) phenotype. For example, murine models with constitutional homozygous MMR gene inactivation have high rates of lymphoma formation, limiting the efficacy of these models. In an effort to circumvent this challenge, investigators have employed tissue-specific Cre recombinase-based inactivation of MMR genes; however, these mice predominantly develop tumors in the small intestine (as opposed to the large intestine in humans) [3]. Although all models bear imperfections, the limitations of cellular cultures and murine models

warrant the need for novel model systems that can contribute to improve the clinical outcomes for both LS and MSI sporadic patients.

Given the anatomic and physiologic similarities and the genomic homology between non-human primates (NHPs) and humans, researchers have used several species to develop therapies and vaccines to treat and eradicate human disease [4,5]. The rhesus macaque (*Macaca mulatta*), which shares 97.5% DNA sequence identity with humans in exons of protein-coding genes as well as close similarity in patterns of gene expression, has been an invaluable animal model for studying human pathophysiology [6,7]. Studies have shown that rhesus launch parallel immune responses to humans, thus making them another available animal model for ascertaining clinical translation of basic and pre-clinical findings in conjunction with other model organisms [8–11].

A cohort of specific pathogen free (SPF), Indian-origin rhesus macaques bred at The University of Texas MD Anderson Cancer Center (MDACC) Michale E. Keeling Center for Comparative Medicine and Research (KCCMR) spontaneously develops MSI/MMRd CRC, including a subset of animals harboring a pathogenic germline mutation in *MLH1* (c.1029C<G, p.Tyr343Ter). This spontaneous mutation manifests into clinical and pathological features analogous to human LS, which suggests that these rhesus macaques may be an informative model organism for studying the biology of MMRd CRC [10,12].

This study characterized the genomic features of colorectal tumors in the KCCMR rhesus cohort using microsatellite marker testing, whole transcriptomics, and epigenomics coupled with systems biology tools, as illustrated in **Fig 1**. Additionally, we cross-compared the current subtypes of CRC in humans with the rhesus model to evaluate the utility of rhesus for studying cancer development, and developing treatment modalities, and prevention approaches in sporadic MSI-H CRC and LS.

## Results

### Clinical characteristics of colorectal tumors in rhesus

We identified a total of 41 animals diagnosed with CRC at the time of necropsy. Specimens were collected between 2008 and 2019. All tumors were located in the right side of the colon (20 in the ascending colon, 16 in the ileocecal valve, and 4 in the cecum) with the exception of one jejunal tumor. The mean age at death was 19.1 years (range: 9 to 27, **Fig 2A**) and 80% of animals were female (**Fig 2B**), consistent with overall population demographics of the cohort from which the animals were drawn. The average age at death was younger among LS animals compared to sporadic MSI-H macaques, but the difference did not reach statistical significant (17.75 vs 19.48 years, *P*-value = 0.2, **S1 Fig**).

### Germline genetics

We detected the presence of a previously described heterozygous germline stop codon mutation in exon 11 of *MLH1* (c.1029C>G; p.Tyr343Ter, **Figs 2C and** S2) in 8 animals (~20%) [10], thus confirming the presence of a causative pathogenic mutation previously described in humans (herein these animals are referred to as rhesus Lynch) [12].The remaining 33 animals (80%) had the wild-type germline sequence of *MLH1* (herein referred to as sporadic rhesus, **Fig 2C**). The pedigree of rhesus Lynch animals revealed the autosomal dominant inheritance pattern of the *MLH1* mutation and an inheritance pattern concordant with the Amsterdam criteria (the 3-2-1 rule) originally described in human LS (**S3 Fig**) [13].

**Immunohistochemistry (IHC) staining displayed widespread loss of expression in MLH1 and PMS2 in rhesus CRC.**   We obtained IHC data from a total of 37 rhesus CRCs (n = 37) with 36 samples (97%) displaying loss of protein expression in MLH1 and/or PMS2.

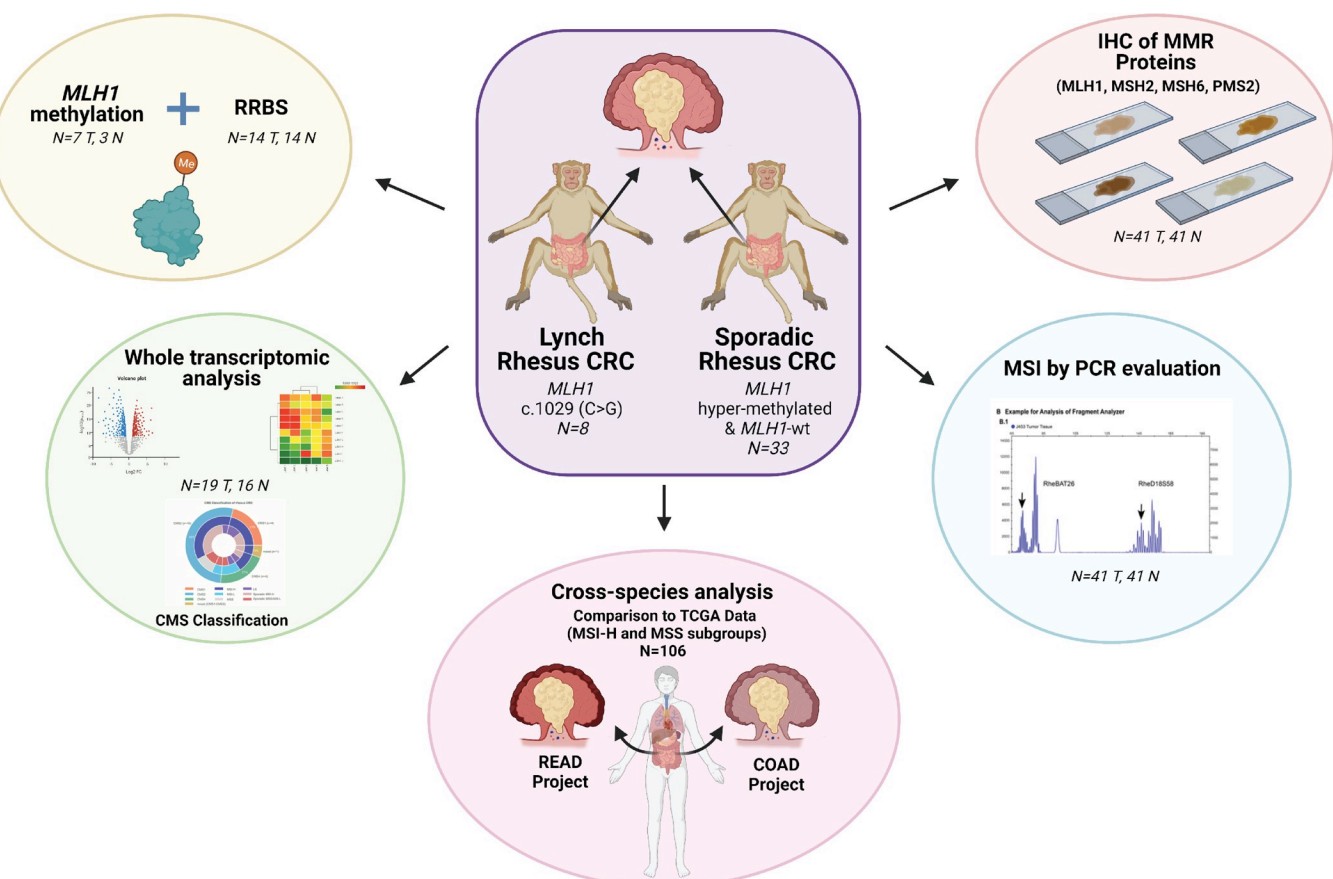

**Fig 1. Schematic outline of the experimental design.** Sporadic and rhesus Lynch (heterozygous *MLH1* nonsense mutation, c.1029, C>G) animals bred and housed at UTMDACC KCCMR were used to genomically characterize colorectal tumors using an in-house MSI marker panel, IHC of MMR proteins, epigenetic assessment, whole transcriptomics analysis, and CMS classification. These analyses established the framework for utilizing rhesus as a surrogate to study MMRd CRC.

Only one animal (~3%) retained the expression of the MLH1-PMS2 heterodimer. This same animal also displayed complete stability of all the MSI markers, thus being MSS, and therefore considered as MMR proficient. Subsequently, we used this animal as a control for all further genomic analyses (**Figs 2D and** S4).

**Assessment of MSI in rhesus CRC.** We developed a PCR-based MSI testing panel for rhesus CRC including orthologs of the most frequently used microsatellite markers in human CRC: BAT25, BAT26, BAT40, D10S197, D18S58, D2S123, D17S250, D5S346, β-catenin, and TGFβRII. Rhesus orthologs of D2S123, D17S250, and D5S346 markers did not contain adequate nucleotide repeats suitable to assess the presence of MSI. Hence, we excluded these markers from our rhesus MSI testing panel. Furthermore, the rhesus ortholog of BAT25 was not sensitive enough to determine MSI due to the interruption of the microsatellite tract by one nucleotide. Therefore, we substituted it with a novel MSI marker (named c-kitRheBAT25), which was identified by screening the whole sequence of *c-kit* to identify an uninterrupted repeat region. Overall, the rhesus CRC MSI testing panel included 6 markers: 4 mononucleotides (c-kitRheBAT25, RheBAT26, RheBAT40, RheTGFβRII) and 2 dinucleotides (RheD18S58, RheD10S197) (S1 Table). This panel offers an assessment of the functionality of the MMR system in these rhesus macaques.

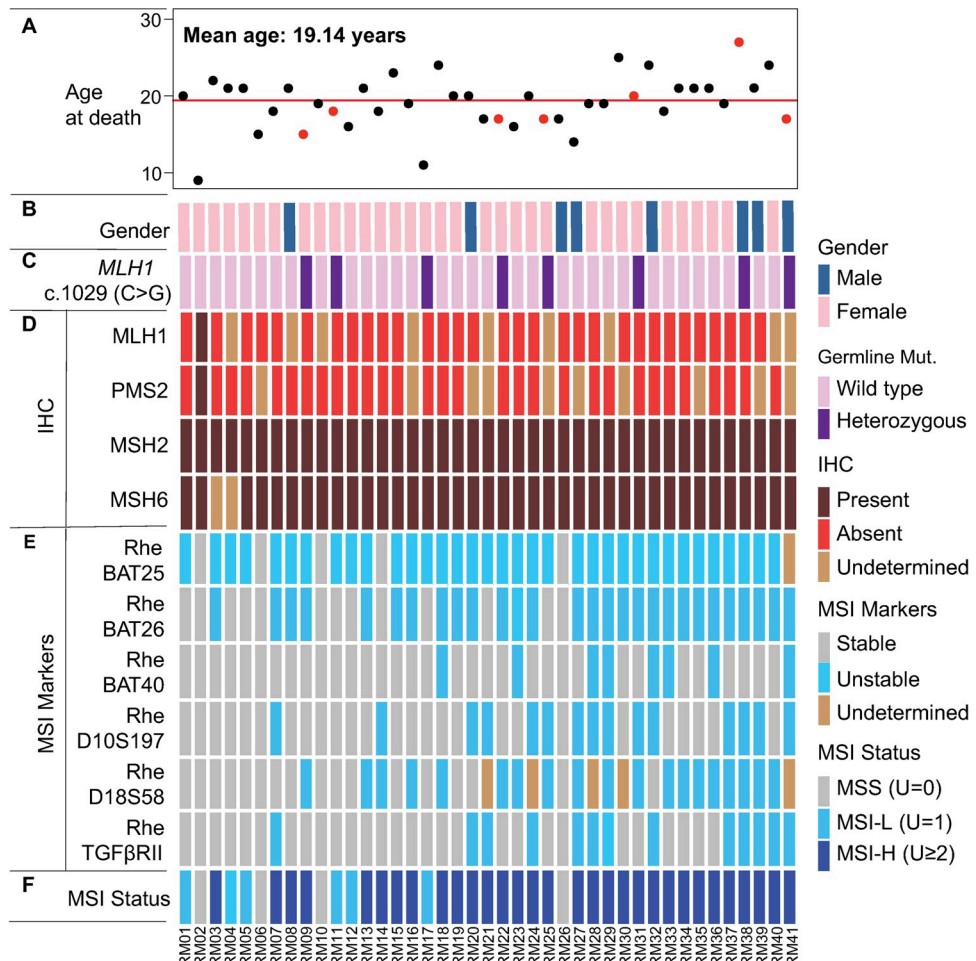

**Fig 2. Clinical, pathological, and molecular characteristics of the Rhesus cohort. (A)** Animal ages at the time of diagnosis of CRC and subsequent euthanasia. The average age at death for the rhesus CRC cohort was 19.14 years. Red dots indicate the age of animals with *MLH1* germline mutation; **(B)** Gender of rhesus cohort. The majority of animals in this cohort were female; **(C)** Lynch syndrome *MLH1* germline mutation status. A total of eight (20%) carried a heterozygous *MLH1* nonsense mutation (c.1029, C>G); **(D)** IHC assessment of rhesus CRC. The majority of rhesus tumor samples displayed loss of MLH1 and PMS2; **(E)** MSI testing of rhesus tumors. Newly designed MSI testing panel for rhesus CRC included six markers (RheBAT25, RheBAT26, RheBAT40, RheD10S197, RheD18S58, and RheTGFβRII) that were orthologs of commonly tested MSI loci in human tumors (BAT25, BAT26, BAT40, D10S197, D18S58, and TGF βRII). Overall, RheBAT25, RheBAT26, and RheD18S58 MSI markers were the most mutated MSI markers in rhesus CRC; **(F)** Summary of MSI status of rhesus tumors. Rhesus CRC were predom-inantly MSI-H (75%), and only six tumors (15%) were MSI-L, and four (10%) MSS.

Using the newly designed rhesus MSI panel, we performed MSI testing in all tumors from the KCCMR cohort using matched normal samples as genomic reference (n = 41). c-kitRhe-BAT25, RheBAT26, and RheD18S58 markers were the most sensitive (**Fig 2E**). We validated the calls made in RheBAT26 and RheD18S58 using an alternative technique based on fragment analysis (**S5 Fig**). We classified rhesus tumors into the three classical categories (i.e. MSI-H, MSI-L, and MSS) by counting the number of unstable markers in each tumor, thus following the classical NCI recommendations [14]. Thirty-one samples were MSI-H (76%), six were MSI-L (15%), and four were MSS (10%, **Fig 2F**). Two rhesus LS animals (RM11 and RM17) displayed an MSI-L phenotype and six rhesus LS animals (RM09, RM22, RM25, RM31, RM38 and RM41) presented an MSI-H phenotype (**Fig 2C and 2F**).

**DNA methylation was responsible for developing CRC in the rhesus.** As seen in human MSI CRC, the MSI testing and the transcriptomic profiling of rhesus MSI-H CRC suggested that the vast majority of rhesus CRC could be secondary to an epigenetic event. To determine the epigenetic contribution to rhesus CRC carcinogenesis, we analyzed global DNA methylation patterns in tumor (n = 14) and matched adjacent normal samples (**S3 Table**). Unsupervised 3D principal component analysis (PCA) of reduced-representation bisulfite sequencing (RRBS) data revealed clear clustering of sporadic MSI-H, MSI-L/MSS as well as normal mucosa with MSI-L/MSS tumors clustering closer to normal mucosa. LS rhesus tumors clustered according to their MSI status with three MSI-H closer to their sporadic counterparts and the MSI-L doing the same with the sporadic MSI-L/MSS group (**Fig 3A**). When we performed

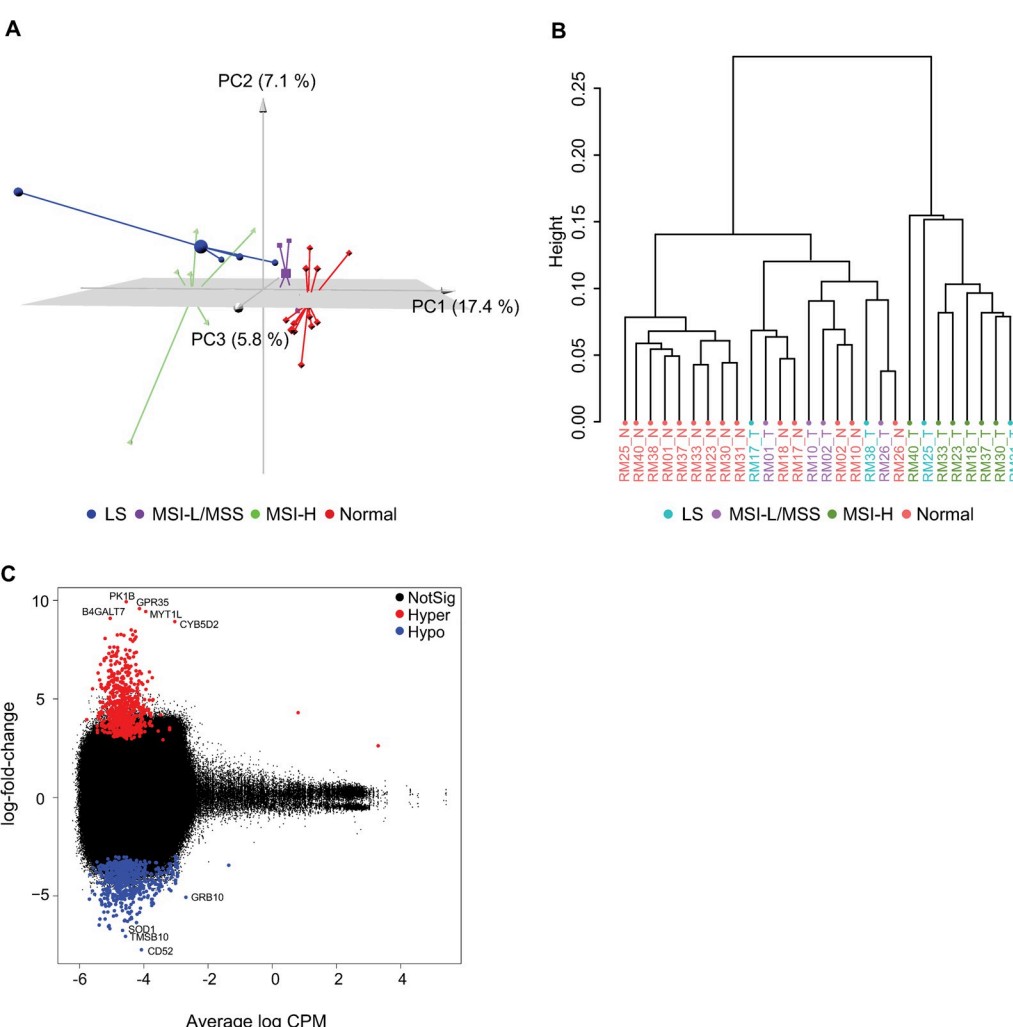

**Fig 3. Methylation analysis of rhesus CRC. (A)** 3D PCA of DNA methylation in rhesus specimens characterizing the trends exhibited by differentially methylated region profiles of sporadic MSI-H (green pyramid), sporadic MSS and MSI-L (purple cube), Lynch syndrome (blue sphere), and normal tissue (red diamond) samples. Each shape represents a tissue sample type. Each group clusters separately; however, sporadic MSS and MSI-L CRC samples are closer to normal tissue samples; **(B)** Hierarchical clustering of DNA methylation profiles assesses by CpG methylation using Pearson's correlation. Distance displays the relationship between rhesus tumors and matched normal tissue samples with parameters set as distance method: "correlation", clustering method: "ward"; **(C)** Significant differentially methylated regions (DMRs) of rhesus tumors displaying MSI-H and MSI-L/MSS phenotypes at FDR of 5%. *PK1B*, *B4GALT7*, *GPR35*, *MYT1L and CYB5D2* are among hyper-methylated genes, and *GRB10*, *SOD1*, *TMSB10* and *CD52* are hypo-methylated.

hierarchical clustering of DNA methylation profiles, a clear separation between sporadic rhesus MSI-H and MSI-L/MSS tumors became evident with normal colorectal samples closer to MSI-L/MSS tumors. The analysis confirmed the robustness of the three main clusters with unbiased *P*-value that were statistically for all the groups (**S6 Fig**) using two methods. Of note, rhesus LS tumors displaying MSI-H patterns (RM25, RM31) clustered together with sporadic MSI-H tumors and the one MSI-L rhesus LS (RM17) was closer to normal and rhesus MSS/MSI-L samples (**Fig 3B**). One rhesus LS animal with MSI-H phenotype (RM38) clustered together with normal and rhesus MSS/MSI-L samples, but the clustering branch of this animal was physically and statistically closer to MSI-H tumors than to MSI-L/MSS and normal tissues (**Figs 3B** and S6). A total of 628 hypermethylated and 592 hypomethylated genes were identified as significant differentially methylated regions (DMRs) using a cut-off of FDR of 5% between the rhesus MSI-H (grouping both sporadic and LS) and MSI-L/MSS (sporadic and one LS tumor) involving some of following genes: *PK1B*, *B4GALT7*, *GPR35*, *MYT1L* and *CYB5D2* (hypermethylated), and *GRB10*, *SOD1*, *TMSB10* and *CD52* (hypomethylated, **Fig 3C**). In addition, we also detected a number of DMRs between rhesus tumor and adjacent normal colorectal mucosa at FDR of 5% (**S7A Fig**). A correlation analysis revealed a negative relation between DNA methylation and gene expression levels that showed a trend towards statistical significance (*P*-value = 0.1336, **S7B Fig**), thus demonstrating that DNA methylation in rhesus CRC affected gene expression levels.

Lastly, we performed a dedicated methylation analysis of the *MLH1* promoter (n = 10) using a methyl NGS panel (**S3 Table**). The location of the CpG regions from the transcription start site of the *MLH1* gene were identified observing thirteen CpGs significantly methylated in rhesus sporadic MSI-H tumor samples compared to adjacent normal mucosa (*P*-value<0.05). The majority of the methylated CpG regions were within exon 1. There were no significant methylation differences between other tumor sub-groups (MSS/MSI-L) and normal tissue samples (**S8 Fig**); however, there was a clear trend of higher levels of *MLH1* promoter methylation among rhesus sporadic MSI-H compared to MSS tumors as well as a notorious absence of *MLH1* methylation in the only LS tumor tested, which is consistent with human CRC biology.

**Gene expression patterns displayed differences between rhesus colorectal tumor and adjacent normal mucosa.** We performed whole transcriptome sequencing in 19 colorectal tumors and 16 matched normal mucosa samples with an average tumor purity estimates in silico of 66% (**S9 Fig**). Unsupervised 3D principal component analysis (PCA) of RNAseq data showed a clear separation between tumor and normal samples. However, samples from the rhesus LS, sporadic MSI-H, and MSS/MSI-L clustered together (**Figs 4A** and S10A). To further characterize the rhesus LS as a model of human MSI-H CRC, we compared rhesus LS tumor to human MSI-H (n = 96) and MSS (n = 440) CRC cases from The Cancer Genome Atlas (TCGA) colon and rectal adenocarcinoma projects (COAD and READ, respectively) using the edgeR package. A total of 101 orthologous genes demonstrated statistically significant changes (BH-adjusted *P*-value < 0.05) in the expression level by at least two-fold difference (log2FC≥1). Then, we aimed to compare global gene expression patterns among rhesus Lynch tumor samples (n = 21) and COADREAD MSI-H and MSS samples to check for their correlation, while using COADREAD (human, n = 54) and rhesus normal samples (n = 20) to control the distance between both species. Rhesus Lynch tumor samples correlated better with COADREAD MSI-H tumor samples (0.82) than that with COADREAD MSS samples (0.68) and normal samples (0.64, **Fig 4B**), thus suggesting that our analysis had sufficient resolution to analyze tumor tissue similarities.

We then determined significantly differentially expressed genes (DEGs) between rhesus normal and tumor using a Benjamini-Hochberg (BH)-adjusted *P*-value≤0.05 and log2 fold

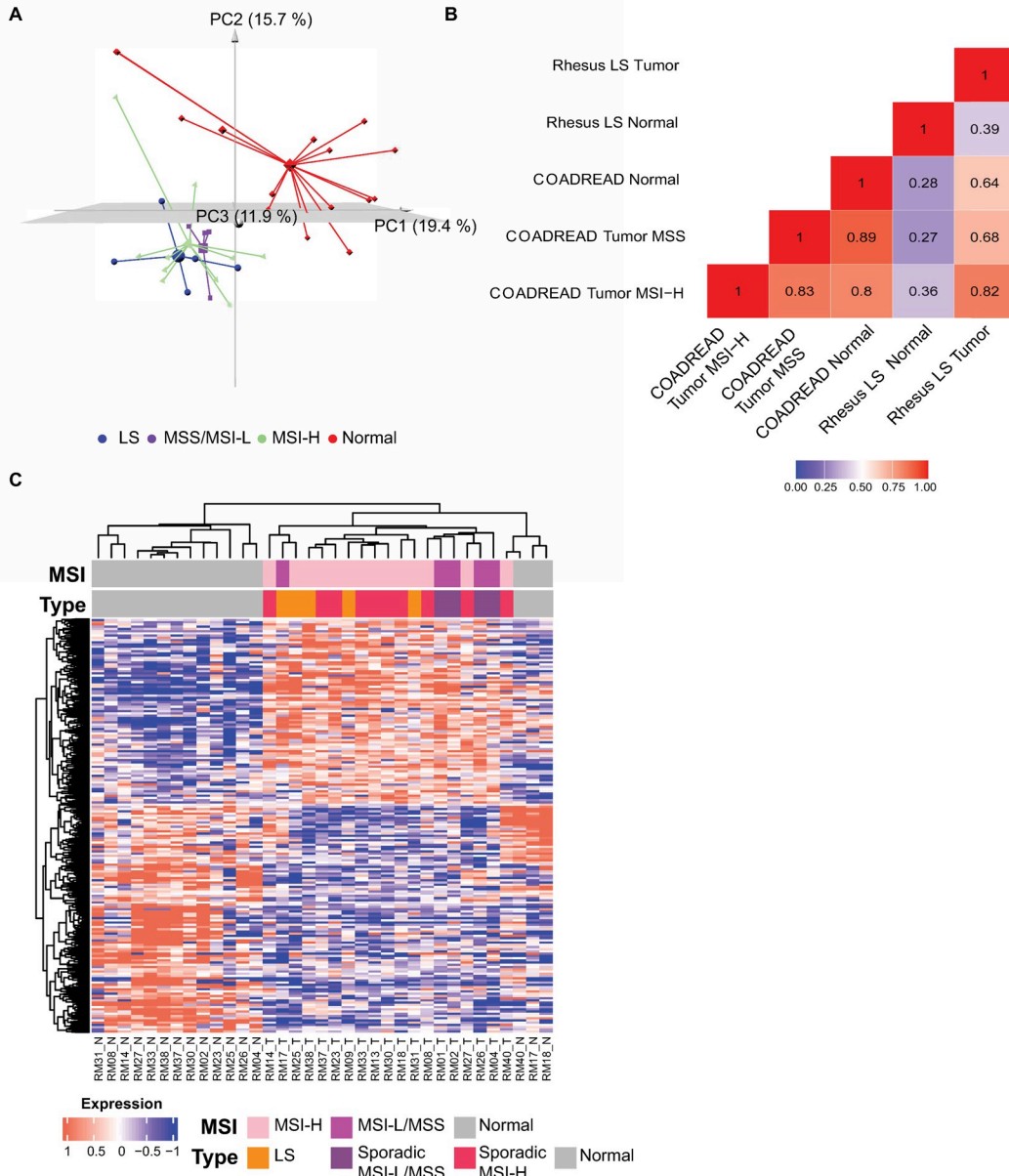

**Fig 4. Transcriptomic analysis of rhesus CRC.** (**A**) 3D principal component analysis (PCA) of rhesus CRC gene expression profiles show clear separation among sporadic MSI-H samples (green pyramids), sporadic MSS and MSI-L (purple spheres), Lynch syndrome (blue cubes), and normal tissue (red diamonds). Normal tissue samples clustered separately from tumor tissue samples; (**B**) Pearson's correlation coefficient of mean expression levels across 101 significant genes from COAD-READ MSI-H tumor samples, COADREAD MSS tumor samples, COADREAD normal tissue samples, rhesus LS tumor samples, and rhesus normal tissue samples; (**C**) Significant differentially expressed genes (DEGs) between tumor and normal tissue samples. DEGs were found based on BH-adjusted *P*-value≤0.05 between rhesus colorectal normal and tumor. Pearson's correlation was used to perform hierarchical clustering between rhesus tumor and normal tissue samples. Columns represent samples, and rows represent statistically significant differentially expressed genes. MSI bar displays MSI status of samples based on PCR-based MSI testing. Gray color represents normal, pink MSI-H, and magenta MSS and MSI-L tissue samples. MSI type bar displays *MLH1* genotyping data with gray color representing normal, orange LS, purple sporadic MSI-L and MSS and red sporadic MSI-H tissue samples.

change±1. We annotated genes using human orthologs (**S10B Fig**). Unsupervised hierarchical clustering using DEGs demonstrated that rhesus tumor tissue samples clustered separately from normal tissue samples, and rhesus MSS/MSI-L CRC were separated from MSI-H CRC samples. Notably, the MSI-L tumor sample from the rhesus LS animal (RM17) clustered with the MSI-H and LS group (**Fig 4C**). Using total RNAseq transcripts, we sought to validate the expression of MMR genes using the read counts from tumors and matched normal samples. *MLH1* read counts in MSI-H CRC samples were significantly decreased compared to normal tissue samples ($P$-value<0.0001). As expected, animal RM02 with a MSS tumor showed more *MLH1* read counts in tumor than matched normal. *MSH6* gene read counts in MSI-H CRC samples were significantly more abundant than matched-normal samples ($P$-value<0.001). Differences of *MSH2* and *PMS2* gene read counts between the rhesus tumor and normal tissue samples were not significant (**S10C Fig**).

Gene set enrichment analysis (GSEA) was performed to discover relevant pathways in colorectal carcinogenesis using the ESTIMATE algorithm, which assesses immune and stromal cell admixtures in tumors, canonical, immune, and metabolic pathways (**Fig 5A–5C**) [15,16]. When compared with normal tissue samples, top pathways enriched in MSI-H tumors were involved in cell cycle regulation, crypt base dynamics, and integrin signaling. Conversely, metabolic pathways in MSI-H samples were downregulated compared to normal tissue (**Fig 5A**). A similar trend was observed for MSS/MSI-L tumor samples compared to normal (**Fig 5B**). Lastly, comparing the significant pathways between MSS/MSI-L and MSI-H, we observed an upregulation of key pathways involved in cell cycle regulation and MYC targeting in the MSI-H group (**Fig 5C**).

**CMS classification categorized rhesus CRC samples mainly as CMS2.** We assigned a consensus molecular subtype (CMS) status to each tumor sample based on the nearest CMS probability (**S3 Table**). Overall, 52% (n = 10) of tumors were classified as CMS2, which corresponds to the canonical pathways of colorectal carcinogenesis; 21% (n = 4) were CMS1, which progresses through MSI and immune pathways; and 21% (n = 4) were CMS4, which develops through mesenchymal pathways. Only one tumor displayed mixed features (CMS1-CMS2) of a transition phenotype (**Fig 5D**).

**Rhesus CRC causes mutations in commonly mutated CRC genes.** We examined somatic variants of rhesus CRC from the total RNAseq data. Our data indicated that mutational rate in coding regions of rhesus CRC was similar in all tested samples (**S11A Fig**). Substitutions of cytosine to thymine were the most abundant somatic variants in rhesus CRC (**S11C Fig**). Commonly altered genes in human CRC were also mutated in rhesus such as *APC*, *ARID1A*, *TGBRII*, *TP53*, *CTNNB1*, *PIK3CA*, *KRAS* (**S11B** and **S12** **Figs**). Due to the close relation found in humans between MSI-H status and *BRAF* mutations, we performed Sanger sequencing to assess the mutational status of the *BRAF* mutation hotspot *V600E* in rhesus CRC. While we did not detect *BRAF V600E* mutations among rhesus tumors, we did observe several types of *BRAF* somatic variants including missense, nonsense, in-frame, and frameshift deletions (**S13 Fig**).

## Discussion

Although cell cultures, organoids, and murine animal models are the most frequently used models in CRC research, these systems sometimes fail to recapitulate all the phenotypic features of MMRd CRC, thus limiting clinical translation to humans. To overcome some of the differences between humans and research models, investigators are also pursuing the use of alternative non-murine models such as dogs, cats, pigs, and NHPs. NHPs are attractive due to their high degree of genomic and physiologic similarity to humans, including natural inter-

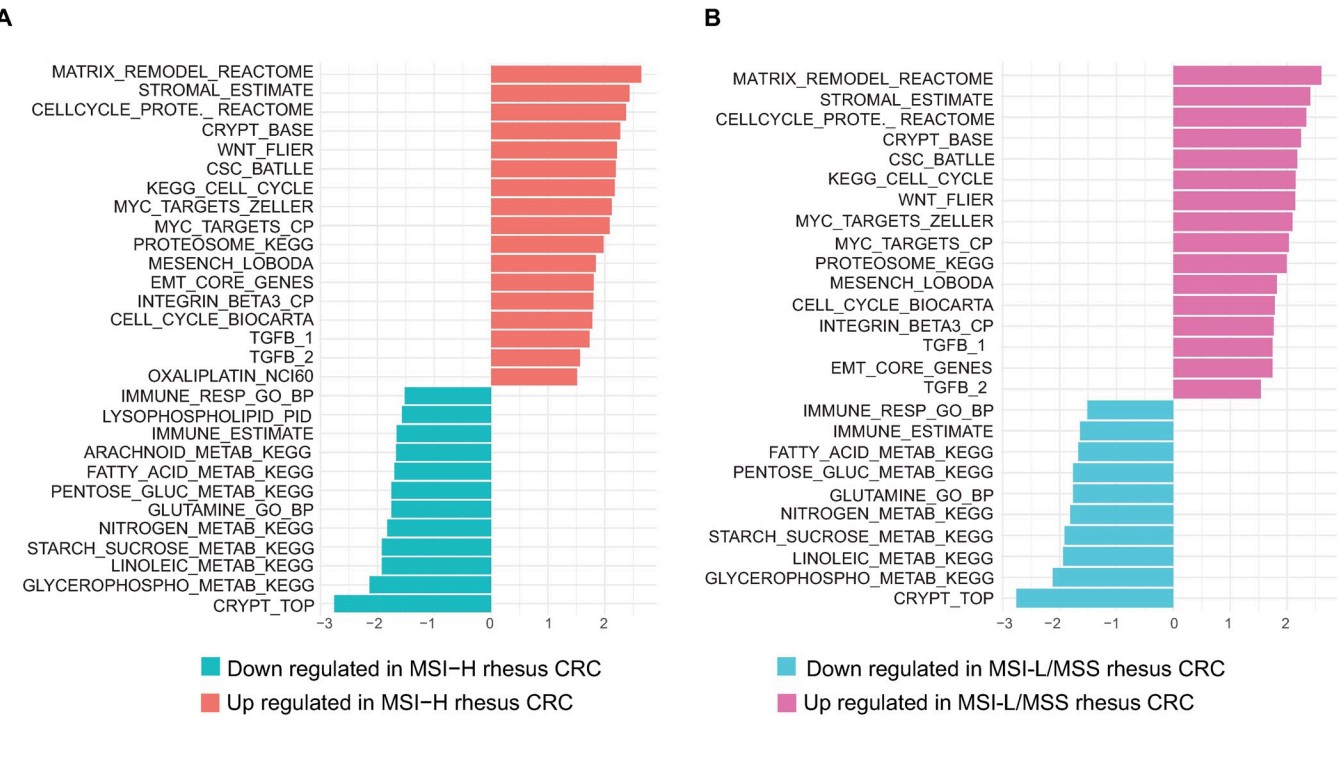

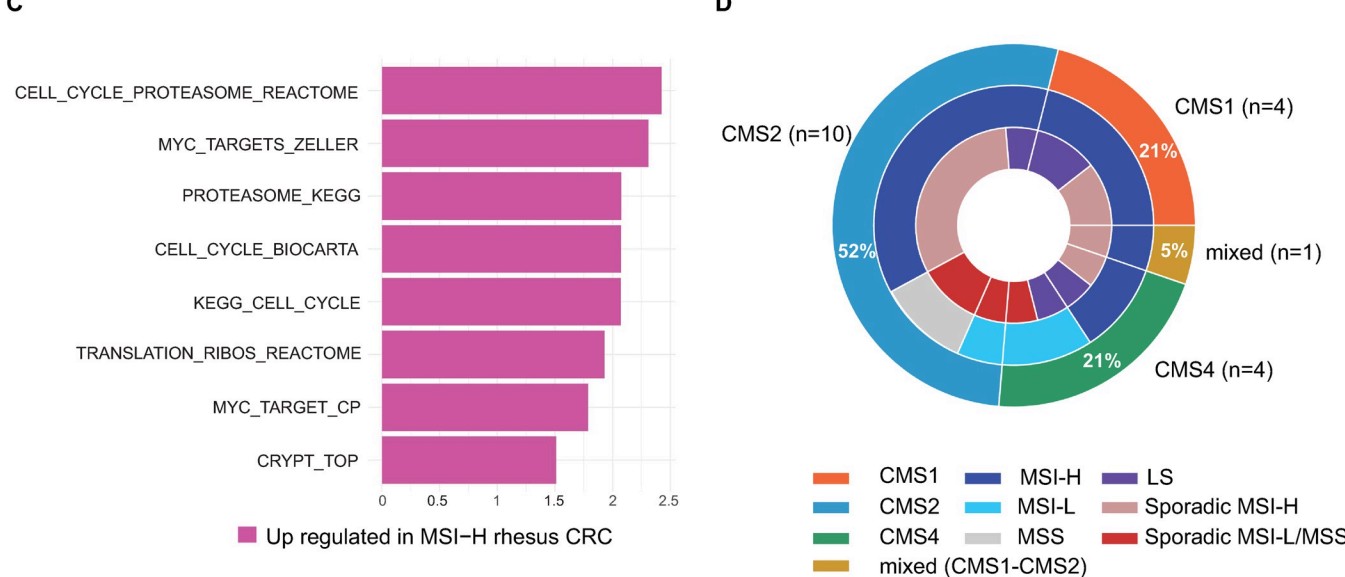

**Fig 5. Gene set enrichment analysis in rhesus CRC. (A-C)** Significant gene expression pathways relevant to CRC biology are highlighted in **(A)** MSI-H (sporadic and LS) and **(B)** in MSI-L/MSS (sporadic and LS) compared to normal tissue samples. **(C)** Highlighted pathways are up up regulated in MSI-H (sporadic and LS) rhesus CRC compared to MSI-L/MSS (sporadic and LS) rhesus CRC. BH-adjusted *P*-value≤0.05 was set as threshold for analysis; **(D)** CMS classification of rhesus CRC. The outer ring of circos plot represents CMS subtypes present in rhesus CRC with 52% of samples (n = 10) classifying as CMS2. Middle ring represents MSI status of samples, and inner ring indicates clinical categories of samples.

individual genetic variation. Previous reports have proven that rhesus macaques serve as durable and clinically relevant animal model to study infectious diseases and cancers [9,10,17,18]. In this study, our results from MSI testing, IHC, gene expression profiling, systems biology approaches, somatic variant calling, and DNA methylation of colon tissue samples from the KCCMR cohort demonstrated that rhesus macaques develop CRC phenotypes analogous to MSI, including both sporadic MSI-H and LS patients. These findings indicate that rhesus macaques may serve as a useful animal model for studying MMRd CRC and address some of the shortcomings of previously established model systems mostly linked to immune-host interactions due to emergence of somatic mutations from MMR deficiency. Nonetheless, it is essential to acknowledge that rhesus, and NHPs in general, have limitations and disadvantages including barriers of high cost, availability, ethical concerns, housing requirements, longer carcinogenic intervals, and intrinsic biological differences.

To characterize the rhesus macaque as a surrogate for studies of MMRd, we investigated the MSI status of 6 markers across 41 unique rhesus tumors using a newly designed, in-house MSI panel for rhesus CRC. Our study results indicated that 76% of rhesus CRC from the KCCMR cohort displayed an MSI-H phenotype, which warrants the use of rhesus as an optimal system to study MMRd carcinogenesis. Many rhesus tumors lost expression of MLH1 and PMS2 proteins, but retained the expression of MSH2 and MSH6, as confirmed by IHC analysis. The *MLH1* germline stop codon mutation (c.1029C>G, p.Tyr343Ter), previously reported as a pathogenic variant in human LS (National Center for Biotechnology Information), was present in 8 (19.5%) rhesus macaques, while the majority (80.5%) were wild-type for this variant. The majority of CRC from *MLH1* mutation carriers presented an MSI-H phenotype; however, two tumors displayed an MSI-L status. This finding is consistent with previous observations in LS patients and reflects that LS carcinogenesis can follow the canonical MMRd route, or a MMRd pathway that is more frequently observed in sporadic carcinogenesis driven by WNT activation [19].

DNA methylation analysis of rhesus CRC suggested that epigenetics plays a pivotal role in the rhesus CRC development. Comparative DNA methylation from colon tumor and adjacent normal tissue samples indicated clear segregation of methylation patterns between MSI-H and MSI-L/MSS CRC samples and also between tumor and adjacent normal mucosa. Interestingly, although human CRC typically displays widespread DNA methylation through the promoter region of the *MLH1* gene, methylation of rhesus CRC predominantly occurred in the exon 1 of *MLH1*.

Despite prior reports of tissue-specific transcriptome analysis of fresh frozen tissues from rhesus macaques, no study had previously profiled the colonic tissue from rhesus macaques of Indian origin [20]. Therefore, this constitutes the first analysis of matched tumor and normal colon samples in rhesus macaques using NGS. Our analysis observed clear expression differences between rhesus tumor and normal samples, and when compared to human CRC data from TCGA, rhesus MSI-H tumors were more similar to human MSI-H than MSS tumors. These findings of transcriptomic similarity between humans and rhesus CRC support utilizing data derived from rhesus LS to study aspects of MMRd carcinogenesis that requires assessment of global transcription patterns such as neoantigen discovery and profiling of CRC. Moreover, to confirm the biological relevance of the rhesus macaque as an animal model, we performed CMS classification and GSEA to ascertain the molecular features of rhesus MSI and LS CRCs. Rhesus CRC mainly associated with CMS2, which is the canonical CRC subtype that corresponds with high levels of copy number changes and activation of WNT/MYC pathways [15]. However, rhesus LS tumors primarily associated with CMS1 (MSI-Immune), which encompasses MSI, CpG Island Methylator Phenotype (CIMP) high, hypermutation, and immune activation, thus aligning with

previous studies from our group [21]. Conversely, most human sporadic CRCs typically cluster with CMS2, which was also observed in a relevant fraction of sporadic rhesus tumor samples. We acknowledge that this observation is not entirely consistent with results from human CRC, but it could reflect that the CMS classifier has been optimized to classify human tumors and it would require computational adaptation to rhesus data to consider intrinsic differences in rhesus tumorigenesis, which is predominantly metastatic. This advanced stage of tumorigenesis at time of detection in rhesus may explain the predominance of CMS2-associated signals, at which point an early-stage CMS1 tumor may have converged toward CMS2 classification due to late-staged WNT activation.

GSEA indicated activation of key pathways—namely cancer stem cell (CSC) signatures and crypt base—in sporadic MSI rhesus CRC, which corroborates a previously described signature of human MMRd CRC [22]. The pathway enrichment between MSI-L/MSS and MSI-H indicates that these advanced, late-stage lesions are transcriptomically similar, which may be driven by the late time point rather than MSI status. These findings provide evidence to support the use of rhesus macaques as a model to understand the molecular basis and tumor micro-environment in MMRd tumorigenesis.

To quantify the mutational rate in rhesus MMRd CRCs, we leveraged RNAseq data of the rhesus LS tissues. This allowed us to observe high mutation rates in genes commonly mutated in CRC, thus adding additional support to the case for utilization of rhesus macaques for vaccine research and immunotherapy development, since there is strong expression of tumor-associated neoantigens derived from observed somatic mutations. In addition, the high average tumor purity in RNAseq (>65%) along with the mean variant allele frequency (VAF) increased the power of our mutation analysis and provided assurance on the level of detection somatic mutations rhesus CRC from RNAseq.

We report a spontaneous NHP model for *MLH1*-mutated Lynch Syndrome and more generally sporadic CRC. An important point to consider in this model is the ability of NHP colonies to maintain a high level of environmental uniformity and patterned genetic relatedness between pedigreed individuals, which affords a deeper understanding of the contributions of genes in complex disease that are not comparably possible with other genetically engineered models or even human populations. Although, gene editing technologies such as CRISPR/ Cas9 seem to have enormous potential to develop novel biomedical research models, NHP models have not been widely used for such gene editing technologies in biomedical research at the present time. In contemplating the future possibility, feasibility, and value of creating targeted gene editing of *MLH1, MSH2, MSH6, PMS*, or *EPCAM* in a NHP with the intent to model MMRd human LS, one should consider the expected length of time for onset for the disease phenotype. Other considerations should include the relative penetrance seen in human LS with each of the MMR genes. MSH6 and PMS2 lead to phenotypes that are less penetrant in humans. MSH2 is as penetrant as MLH1 in humans, but more frequently leads to ovarian or endometrial cancer. Such considerations should be balanced *a priori* against what questions would be expected to be answered with a gene-edited NHP that could not be potentially answered by a spontaneous model.

We acknowledge that our study has several limitations necessitating further investigation. Importantly, the comparator group, MMR proficient (MMRp) tumors, only included one animal, which challenged the validity of the comparison between MMR proficiency and deficiency. Thus, a stronger comparator group is necessary to strengthen our findings. Furthermore, this study lacks pertinent information regarding the timeline of carcinogenesis for both sporadic and LS rhesus tumors, which restricts our understanding of pre-cancer biology, and the timing of tumor development and subsequent evolution. In addition, neoantigen detection and T-cell receptor (TCR) profiling would be an important asset for a complete

understanding of the immune system in the rhesus macaque CRC. Lastly, our mutation calling was performed using total RNA sequencing data, which, although adequate, is less ideal than whole exome sequencing.

In conclusion, this study provides a robust molecular and genetic characterization of a spontaneous and translationally relevant NHP animal model that will be useful for understanding MMRd CRC, including LS CRC. These results justify the preclinical use of the rhesus to study LS CRC and also the larger group of sporadic MSI CRCs in specific contexts, such as survey the immune landscape, discovery of prevention strategies, assessment of TME dynamics, development of treatment towards advanced cancers, all in a model system with moderate to high translational value to humans.

## Material and methods

### Ethics statement

All animal experiments were approved by the institutional animal care and use committee (IACUC) and the care of the animals was in accordance with institutional guidelines (IACUC protocol #0804-RN02). Animal care and husbandry conformed to practices established by the Association for the Assessment and Accreditation of Laboratory Animal Care (AAALAC), The Guide for the Care and Use of Laboratory Animals, and the Animal Welfare Act.

### Animal care

The rhesus macaque colony detailed in this manuscript is housed and maintained at MDACC KCCMR in Bastrop, TX. The breeding colony of Indian-origin rhesus macaques (*Macaca mulatta*) at KCCMR is a closed breeding colony, which is specific pathogen free (SPF) for Macacine herpesvirus-1 (Herpes B), Simian retroviruses (SRV-1, SRV-2, SIV, and STLV-1), and *Mycobacterium tuberculosis* complex.Tissue specimens from the proximal colon (n = 20), the ileocecal junction (n = 16), cecocolic junction (n = 2), cecum (n = 2), and jejunum (n = 1), as well as blood samples of rhesus macaques, were collected opportunistically at necropsy following euthanasia for clinical reasons between 2008 and 2019. Formalin-fixed paraffin-embedded (FFPE) blocks and hematoxylin and eosin (H&E) slides were prepared by veterinary pathology technicians and the diagnosis confirmed by veterinary (C.L.H.) and human pathologists (M.W.T) after necropsy procedure.

### Colony demographics

The rhesus macaque breeding colony, called the Rhesus Monkey Breeding Research Resource (RMBRR), was established as a SPF colony in 1989. Establishment of the colony began in 1974–1975 with 17 male and 74 female Indian-origin rhesus macaques. The colony has remained SPF since 1989 and has been closed since its founding. Mating schema is harem breeding with one male and generally 3 to 12 females in a social breeding group. Kinship coefficients of all males and females are screened prior to assembling breeding groups to minimize inbreeding. Current practice is to avoid mating between animals with kinship coefficients >0.007. For reference, two $2^{nd}$ cousins have a kinship coefficient of 0.0156, and two $3^{rd}$ cousins have 0.00391. Environmental conditions, including diet and behavioral enrichment are comparable across all animals in the colony and have not changed significantly over time.

The first case of CRC in the RMBRR was diagnosed 1988 in one of the founding males. The exact age of this animal is not known but estimated to be >15 years. The allele frequency of the *MLH1* mutation within the RMBRR is currently approximately 5%. *MLH1* carriers have not been segregate in the population. Cumulative prevalence of CRC in the RMBRR in the

period 2003–2019 is: 0.11% for ages 8–12 years; 1.68% for 13–17 years; 5.22% for 18–22 years, and 10.17% for 23–27 years. Rhesus *MLH1* carriers with CRC reported in this paper demonstrate inheritance patterns consistent with autosomal dominance with incomplete penetrance (**S3 Fig**).

### Nucleic acid extraction

Macro-dissection was performed to decrease the admixture of adjacent normal tissue and to enrich the percentage of tumor material for subsequent DNA and RNA extraction. De-paraffinization of FFPE tumor and adjacent normal specimens was performed using QIAGEN de-paraffinization solution (QIAGEN, Valencia, CA). DNA and RNA from 39 tumor and adjacent normal samples were extracted using the AllPrep DNA/RNA FFPE Kit (QIAGEN) following the manufacturer's protocol. In the case of the unavailability of FFPE samples, genomic DNA and RNA were extracted from fresh frozen tumor (n = 2) and normal (n = 3) samples using the ZR-Duet DNA/RNA MiniPrep extraction kit (ZYMO Research, Irvine, CA). Quantification was performed using a NanoDrop One spectrophotometer (Thermo Fisher Scientific, Waltham, MA) and Qubit Fluorometer 2.0 (Qubit, San Francisco, CA) using dsDNA and RNA assay kits. RNA integrity was analyzed using the Tape Station RNA assay kit (Agilent Technologies, Santa Clara, CA). Extracted DNA and RNA were kept at –20 and –80˚C.

### Panel design for MSI testing

Commonly used human MSI markers (BAT25, BAT26, BAT40, D10S197, D18S58, D2S123, D17S250, D5S346, β-catenin, and TGFβRII) were used as a reference to design a panel of rhesus MSI markers [23,24]. In brief, genomic positions of human MSI markers in the rhesus macaque genome (rheMac8) were identified using the batch coordinate conversion tool (lift-Over) in the UCSC genome browser [25]. Repeat patterns were compared to human MSI markers (**S1 Table**). Orthologous microsatellite regions corresponding to human MSI markers D2S123, D17S250, and D5S346 were not specific to assess MSI in the rhesus genome. Therefore, they were excluded from the final MSI rhesus panel. Primer sequences to target identified microsatellite regions in rhesus were designed using the NCBI Primer Blast tool (Accession ID# GCF_000772875.2) [26]. The primer efficiency was evaluated using the UCSC Genome Browser In-Silico PCR tool [25] with rheMac8 as a reference control. The Baylor College of Medicine genome database was used to calculate the probability of encountering SNPs within the primer sequences. Primers sequences with allele frequency greater than 0.05% were redesigned (**S2 Table**).

### PCR-based MSI testing in rhesus CRC

Multiplex PCRs were designed with at least 25 bp size differences among PCR amplicons to afford clear distinction and identification on electropherograms from the Agilent Bioanalyzer 2100. All markers were amplified in 25 μL PCR reactions using 12.5 μL of AmpliTaq Gold 360 PCR master mix (Thermo Fisher Scientific, Waltham, MA), corresponding primer sets, and 10 ng of FFPE DNA. Multiplex PCRs were performed in a Veriti 96 Well Thermal Cycler (Applied Biosystems, Foster City, CA) under the following cycling conditions: initial denaturation at 95˚C for 10 min, followed by 35 cycles at 95˚C for 30 sec, 55˚C for 30 sec, and 72˚C for 30 sec. A final extension at 70˚C for 30 min was implemented to aid non-template adenine addition. Multiplex PCR products were resolved on a 5% ethidium bromide-stained agarose gel. Multiplex PCRs were analyzed via Agilent 2100 Bioanalyzer DNA 1000 kit (Agilent Technologies, Santa Clara, CA). Electropherograms of adjacent normal and tumor tissue samples were compared to assess the status for each of the MSI markers. Following NCI MSI testing

consensus guidelines, MSI status was assigned by counting the number of unstable MSI markers and samples were assigned to either: MSS (stable markers), MSI-L (1 unstable marker, $\leq$ 30%), or MSI-H (2 or more unstable markers, $\geq$ 30%) [14].

## MSI testing via fragment analysis for validation of the RheBAT26 and RheD18S58 markers

Fragment analysis (Applied Biosystems, Foster City, CA) was performed to validate MSI results from the Agilent 2100 Bioanalyzer for RheBAT26 and RheD18S58 MSI markers. In brief, the 5' end of the forward primer sequences for RheBAT26 and RheD18S58 MSI markers was labeled with a 6-FAM fluorescent dye (Thermo Fisher Scientific, Waltham, MA). A multiplex PCR was designed to amplify RheBAT26 and RheD18S58 MSI markers with labeled primer sequences. PCR master mix and conditions were adopted from well-established PCR experiments. The fragment analysis method was performed by the Advanced Technology Genomics Core at MDACC.

## Sanger sequencing for discovery of germline MLH1 and somatic BRAF mutations

Primer sequences were designed to target *de novo* stop codon *MLH1* and *BRAF* mutations following previously described procedures (see panel design section, **S2 Table**). PCRs were performed using the Veriti 96 Well Thermal Cycler (Applied Biosystems, Foster City, CA) under the following cycling conditions: initial denaturation at 95˚C for 10 min, followed by 35 cycles at 95˚C for 30 sec, 55˚C for 30 sec and 72˚C for 30 sec, with a final extension at 72˚C for 7 min. Purification of PCR products was performed with an in-house ExoSAP solution [50 μL of Exonuclease I (20,000 units/mL, NEB M0568, Ipswich, MA); 40 μL of Antarctic Phosphatase (5,000 units/ml); 16 μL of Antarctic Phosphatase buffer (NEB M0289S, Ipswich, MA); 144 μL of nuclease-free $H_2O$]. PCR conditions for purification of PCR products were incubation at 37˚C for 15 min and at 80˚C for 15 min. Quality control of PCR products and purified PCR products was performed running 1% Agarose gel prepared with 25 ml of 1X TBE buffer and 1.2 μL of EtBr. Then, gel-purified PCR products were sequenced by the MDACC sequencing core (ATGC) via the Sanger Sequencing method. Analysis of Sanger sequencing data was performed using DNASTAR lasergene software.

## Immunohistochemistry (IHC)

Immunohistochemistry (IHC) staining for MLH1, MSH2, MSH6, and PMS2 was performed in FFPE tissue sections. Tissue sections were cut at 4 μm and submitted to the MDACC Research Histology, Pathology, and Imaging Core (RHPI) in Smithville, TX. The following Agilent Dako IHC antibodies were used according to manufacturer's recommendations: IR079, Monoclonal Mouse Anti-human Mutl Protein Homolog 1, clone ES05 for MLH1; IR085, Monoclonal Mouse Anti-human Muts Protein Homolog 2, clone FE11 for MSH2; IR086 Monoclonal Rabbit Anti-human Muts Protein Homolog 6, clone EP49 for MSH6; IR087, Monoclonal Rabbit Anti-Human Posteiotic Segregation Increase 2, clone EPS1 for PMS2 [10].

## Total RNA sequencing

Truseq stranded total RNA library preparation kit (Illumina, San Diego, CA) was used to prepare libraries of 19 tumors and 16 matched normal RNA samples, which were extracted from FFPE and frozen tissue samples. Prepared libraries were sequenced for 76nt paired-end sequencing on HiSeq4000 and NovaSeq6000 sequencers (Illumina, San Diego, CA) (**S4 Table**).

## Assessment of DNA methylation testing of *MLH1*

DNA methylation analysis of the *MLH1* gene was performed on DNA from frozen tissue samples of 7 tumors and 3 normal tissue (duodenum and blood) samples using a targeted NGS assay (EpigenDx, Hopkinton, MA) (**S4 Table**). In brief, the bisulfite-treated DNA samples were used as a template for PCR to amplify a short amplicon of 300–500 bp using a set of primers that cover the *MLH1* genomic sequence at -4 kb to + 1kb from the transcriptional start site (TSS). Later, methylation libraries were constructed for methylation analysis on the Ion Torrent instrument at EpigenDx.

## DNA methylation assessment via reduced representation bisulfite sequencing (RRBS)

DNA libraries of RRBS were constructed from FFPE tissue samples of 14 tumors/adjacent normal tissue pairs using the Ovation RRBS Methyl-Seq System at The Epigenomics Profiling Core (EpiCore) of MDACC (**S4 Table**). In preparation, DNA was digested with a restriction enzyme and selected for size based on established protocols used in the EpiCore. Post-adapter ligation ensured enrichment for CpG islands, and DNA was bisulfite-treated, amplified with universal primers, and qualified libraries were then sequenced on Novaseq6000 sequencer at the UTMDACC ATGC.

## Bioinformatics analysis

The FASTQC toolkit was performed for quality control of FASTQ files generated from RNA sequencing [27]. The fastp tool was performed to trim adapters and low-quality reads [28]. Fasta and gtf files of the reference genome (Mmul_8.0.1) were downloaded from the Ensembl genome browser [29]. The reference genome was indexed using the STAR RNA sequencing aligner. Cleaned reads of total RNA sequencing were aligned to the reference genome using the STAR RNA sequencing aligner. Gene level estimated read counts were calculated by STAR RNA sequencing aligner and were saved in reads per gene tabular files [30]. This pipeline was implemented on the high-performance computing (HPC) cluster of MDACC. As performed for total RNA sequencing, RRBS FASTQ files were quality controlled using the FASTQC toolkit [27]. TrimGalore was performed to trim adapters and low-quality reads. Diversity trimming and filtering were completed with NuGEN's diversity trimming scripts. Processed fastq files were aligned to the reference genome (Mmul_10) with bismark bisulfite mapper. The methylation information was extracted with bismark methylation extractor script.

Count data per sample was generated by STAR RNA sequencing aligner and combined into one matrix for downstream analyses. Genes with less than a sum of 300 reads in all samples were excluded from the analysis. Estimated read counts were normalized with variance stabilizing transformation (VST) using the DESeq2 Bioconductor R package [23,31–33]. MSI-L and MSS CRC cases were combined based on previous human studies. 3D Principal component analysis (PCA) of RNA sequencing was performed using pca3d (version 0.10.2) package in R (version 4.0.0) The batch effect of RNA seq data was removed using limma:removebatcheffect package in R (version 4.0.0) [34]. Significant differentially expressed genes between MSI-H and MSS/MSI-L rhesus CRC were calculated using Benjamini-Hochberg (BH)-adjusted *P*-value≤ 0.05 and log2 fold change ≥-1 and log2 fold change ≤1. Unsupervised hierarchical clustering was performed via Pearson's correlation. Comparisons of MMR gene counts between tumor and adjacent normal colorectal mucosa were performed using the DESeq2 Bioconductor R package. Complex heatmap and an enhanced volcano plot were created in R studio (version 3.6.1) [35]. Rhesus Ensembl gene-IDs were converted to human

Entrez ID for the CMS classification and GSEA. CMS classification of tumor samples was predicted using the random forest (RF) predictor in CMSclassifier R package (version 3.6.1) [15, 21]. CMS classification was assigned to the subtype with the highest posterior probability. GSEA was performed with 1,000 permutations using CRC pathways with the fgsea R package [15,36]. CRC pathways included signatures of interest in CRC, the ESTIMATE algorithm that assesses immune and stromal cell admixture in tumor samples, canonical pathways, immune signatures, and metabolic pathways [16,36].

Somatic and germline variant analyses of rhesus CRC samples were performed following GATK best practices. Filtered variants by the Mutect2 tool of GATK were annotated with Variant Effect Predictor (VEP) [37]. Variants with less than 10 reads were excluded. Mutation rates were calculated by dividing the number of non-synonymous somatic mutations in coding regions by the number of callable bases. Callable bases are calculated with samtools using tumor bam files [38]. Tumor purity of samples were calculated using ISOpureR (version 1.1.3) package using tumor and normal gene expression data [39].

Species comparison using TCGA datasets utilized raw RNA-Seq counts of MSI-H and MSS colorectal tumor samples and corresponding normal tissue samples (the 2016-01-28 analyses) of the TCGA project COADREAD and MSI status information was downloaded via FirebrowseR (version 1.1.35) package [40,41]. Then the raw data was filtered (min.count = 10, min. total.count = 15, large.n = 10, min.prop = 0.7) and normalized (TMM method) by package edgeR (version 3.32.1) [42]. Genes showing statistically significant (BH-adjusted p-value < 0.05) changes in the expression level by at least two-fold (log2FC = 1) between MSI-H and MSS samples were identified for the following analysis. The rhesus homologs were found by the Ensembl genome database via the biomaRt package (version 2.46.3) [29,43,44]. Mean CPM (counts per million) of each in COADREAD MSI-H tumor tissues, COADREAD MSS tumor tissues, COADREAD normal tissues, rhesus LS tumor tissues, and rhesus LS normal tissues were used to calculate the Pearson's correlation of each group. CPM of each gene was used to perform the unsupervised hierarchical clustering, and to generate the dendrogram tree and heat map for individual samples.

For DNA methylation analysis of RRBS, 3D PCA was performed using pca3d package in R (version 4.0.0) and sample clustering was performed using cytosine report files pvclust package in R (version 4.0.0 with 10,000 bootstraps [45]. The minimum coverage depth was 10 reads. DMR were calculated using bismark coverage report files with edgeR Bioconductor R package [28]. Significant DMRs at CpG loci were displayed at an FDR of 5%.

Correlation analysis between methylation and expression data was performed using top 500 significant up- and down-regulated genes that were also methylated from the comparison between rhesus tumor and normal adjacent colorectal mucosa. *P*-values were calculated using one-tailed t-test.

## Supporting information

**S1 Fig. Comparison of mean age at death of rhesus presenting with CRC.** Sporadic animals do not carry a germline mutation in *MLH1*. Mean age of CRC at death among rhesus Lynch was younger (17.75 years) compared to sporadic rhesus (19.48 years). Welch's t-test, *P*-value = 0.2169.
(EPS)

**S2 Fig. *MLH1* germline mutation detected in rhesus LS.** Each colored line represents a different type of nucleotide. Brown arrowhead points to the germline mutation detected in normal tissue of rhesus RM09. Non-syndromic animal DNA carries a cytosine (C) nucleotide in c.1029 position of *MLH1*. However, rhesus RM09 carries a mutation in one allele involving the

substitution of C>G in c.1029, thus creating a nonsense mutation that leads to a stop codon (TAG).
(EPS)

**S3 Fig. Pedigree of rhesus cases characterized in this manuscript.** Red marks indicate CRC. Blue mark indicates salivary tumor. Plus signs indicate the presence of the *MLH1* germline mutation in heterozygous state. Generations are indicated using roman numerals on the right margin. Animal U was genotyped and found to be wild type for the *MLH1* germline mutation (c.1029 C>G). All animals are deceased, except animal U.
(EPS)

**S4 Fig. Immunohistochemical staining for MLH1, MSH2, MSH6 and PMS2 from rhesus CRC tissue samples.** Left column indicates IHC results of the colonic epithelium of unaffected rhesus. Middle column shows the IHC results of RM02 that displayed a MSS phenotype retaining the expression of all MMR proteins. Right column displays the results of IHC in RM32 consistent with MSI-H phenotype with loss of expression of MLH1 and PMS2. Note that the internal positive control in case RM32 is the positive expression of the MMR proteins in the lymphocytes present in the stroma. Magnification is 100X.
(EPS)

**S5 Fig. Microsatellite instability analysis. (A)** Examples of microsatellite loci analyzed using the Agilent 2100 Bioanalyzer. Blue and red lines represent tumor tissue and normal tissue, respectively. Electropherograms A1, A2, and A3 are examples of the most frequent microsatellite markers displaying instability in the tumor samples. Arrowheads indicate instability in MSI markers. Electropherograms A4, A5, and A6 are examples of the most common microsatellite markers displaying stability in tumor samples; **(B)** Examples of microsatellite loci analyzed using fragment analyzer. B1 shows tumor tissue of case RM13, displays instability in markers RheBAT26 and RheD18S58. Arrowheads indicate unstable markers in tumor tissue. B2 displays normal tissue of case RM13 and serves as a control/reference to establish calls in microsatellite markers in matched tumor tissue of the same case. Overall, fragment analysis validates the results obtained from Agilent 2100 Bioanalyzer calls in markers RheBAT26 and RheD18S58.
(EPS)

**S6 Fig. Cluster analysis of RRBS DNA methylation analysis using "correlation" distance with"ward.D" clustering method with 10000 bootstrap.** Gray values represent the rank of the cluster and the highest rank is 26 for this cluster. Red values indicate approximately unbiased (au) *P*-value and green values show bootstrap probability (bp). The minimum au value is 96, which proves that the clusters are valid (*P*-value<0.05).
(EPS)

**S7 Fig. DNA methylation regulates gene expression in rhesus CRC. (A)** Significant differentially methylated regions (DMRs) at FDR of 5% observed in rhesus tumors compared to adjacent normal colorectal mucosa. *TOP1*, *PCGF3* and *FAM76B* were among the hyper-methylated genes, and *GAS8*, *ALKBH5* and *MME* were among the hypo-methylated genes in rhesus CRC. **(B)** Correlation analysis between DNA methylation and gene expression data. We observed a negative correlation between DNA methylation and gene expression that had a trend towards statistical significance (*P*-value = 0.1336). For this analysis, we selected the top 500 differentially methylated genes.
(EPS)

**S8 Fig. DNA methylation in the promoter region of *MLH1* in rhesus CRC.** The location of CpG islands are shown from the TSS of *MLH1*. A total of thirteen CpG regions are significantly methylated in sporadic MSI-H rhesus CRC samples (*\*P*-value<0.05 Wilcoxon Signed Rank Test). The majority of methylated CpG regions are located in exon 1 of *MLH1* of rhesus tumor. There is no significant differences in methylation between tumor and normal samples. (EPS)

**S9 Fig. Rhesus tumor purity of specimens assessed by RNA sequencing.** The mean of rhesus tumor purity is 65.9% measured in silico using RNA sequencing data. (EPS)

**S10 Fig. Expression data of rhesus CRC. (A)** 3D PCA of rhesus CRC expression profiles without batch effect correction. **(B)** Differentially expressed genes between rhesus colorectal normal and tumor samples. Gene expression is displayed in volcano plots with log2(FoldChange) on the X-axis and -log10(BH-adjusted *P*-value) on the Y-axis. The horizontal dash line represents BH-adjusted *P*-value = 0.05. The left and right vertical lines represent log2(Fold-Change) = ±1. Significant genes are labeled as upregulated (red) and down-regulated (blue) genes. Some significant genes are annotated; **(C)** Expression of MMR genes in rhesus CRC. Normalized gene counts of whole transcriptome sequencing with variance stabilizing transformation (VST) are on the Y-axis to display gene expression differences of *MLH1*, *MSH6*, *MSH2*, and *PMS2* genes between matched tumor and normal tissue samples. *MLH1* gene expression was significantly (*\*\*\*\*P*-value< 0.0001) low in MSI-H tumor tissue samples, while *MSH6* gene expression was significantly (*\*\*\*P*-value<0.001) higher in MSI-H tumors compared to matched adjacent normal. RM02_T (green star) is the only CRC case with a higher expression of *MLH1* in tumors compared to the matched adjacent tissue sample. (EPS)

**S11 Fig. Analysis of somatic variants in rhesus CRC. (A)** Nonsynonymous mutation rate in coding regions is expressed as mutations per megabase (Mb); **(B)** Commonly mutated genes in human CRC are also altered in rhesus CRC. Each color represents different somatic variants reflected in the figure legend. Black represents multi-hit variants (more than one somatic alteration in that gene); **(C)** Proportions of base-pair substitutions in somatic variants in rhesus CRC. Each color demonstrates a different substitution type with C>T being the most abundant in rhesus CRC. (EPS)

**S12 Fig. Variant allele frequencies of commonly mutated human CRC genes in rhesus CRC.** (EPS)

**S13 Fig. Somatic mutations in BRAF.** Missense, nonsense, in-frame deletion and frameshift deletion mutations detected in BRAF. No mutation hotspots were detected. (EPS)

**S1 Table. Comparison of human and rhesus MSI markers.** (PDF)

**S2 Table. Primer sequences for determination of rhesus MSI status and *MLH1* germline mutation.** (PDF)

**S3 Table. CMS classification of rhesus CRC.** This analysis was performed using a random forest classifier in the CMSclassifier (R studio) to establish CMS status in rhesus CRC. CMS

calls are indicated for each of the samples in bold.
(PDF)

**S4 Table. Summary of genomic analyses performed in specimens from *Rhesus macaques* presented in this manuscript.** Total RNA sequencing was performed in 19 tumor and 16 matched normal samples. DNA methylation using RRBS was performed in 14 tumor and 14 matched normal tissues. Methylation analysis of *MLH1* was performed in 7 tumor and 3 normal samples (2 matched). Samples profiled with more than one platform are marked with a blue background.
(PDF)

## Acknowledgments

We acknowledge the support of Dr. Awdhesh Kalia at the School of Health Professions of MDACC for providing access to the Agilent 2100 Bioanalyzer for MSI testing analysis. We acknowledge the support of the Advanced Technology Genomics Core (ATGC) for performing the RNAseq, Sanger sequencing, fragment analysis, and RRBS of this project; and Dr. Marcos R. Estecio for RRBS library preparation; and support of the High-Performance Computing facility, which provided computational resources.

## Author Contributions

**Conceptualization:** Stanton B. Gray, R. Alan Harris, Krishna M. Sinha, Jeffrey Rogers, Eduardo Vilar.

**Data curation:** Nejla Ozirmak Lermi, Charles M. Bowen, Laura Reyes-Uribe.

**Formal analysis:** Nejla Ozirmak Lermi, Beth K. Dray, Nan Deng, R. Alan Harris, Muthuswamy Raveendran, Fernando Benavides, Carolyn L. Hodo, Melissa W. Taggart, Jeffrey Rogers.

**Funding acquisition:** Stanton B. Gray, Jeffrey Rogers, Eduardo Vilar.

**Investigation:** Nejla Ozirmak Lermi.

**Methodology:** Nejla Ozirmak Lermi, Eduardo Vilar.

**Project administration:** Eduardo Vilar.

**Resources:** Stanton B. Gray, Krishna M. Sinha, Jeffrey Rogers, Eduardo Vilar.

**Software:** Nan Deng, R. Alan Harris, Muthuswamy Raveendran.

**Supervision:** Stanton B. Gray, Krishna M. Sinha, Jeffrey Rogers, Eduardo Vilar.

**Writing – original draft:** Nejla Ozirmak Lermi, Stanton B. Gray, Charles M. Bowen, Laura Reyes-Uribe, Krishna M. Sinha, Jeffrey Rogers.

**Writing – review & editing:** Nejla Ozirmak Lermi, Stanton B. Gray, Charles M. Bowen, Laura Reyes-Uribe, Beth K. Dray, Nan Deng, R. Alan Harris, Muthuswamy Raveendran, Fernando Benavides, Carolyn L. Hodo, Melissa W. Taggart, Karen Colbert Maresso, Krishna M. Sinha, Jeffrey Rogers, Eduardo Vilar.

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
