## [Decision Letter · Decision Letter 0]

24 Oct 2021

Dear Dr Vilar,

Thank you very much for submitting your Research Article entitled 'Comparative Molecular Genomic Analyses of a Spontaneous Rhesus Macaque Model of Mismatch Repair-Deficient Colorectal Cancer' to PLOS Genetics.

The manuscript was fully evaluated at the editorial level and by independent peer reviewers. The reviewers appreciated the attention to an important problem, but raised some substantial concerns about the current manuscript. Based on the reviews, we will not be able to accept this version of the manuscript, but we would be willing to review a much-revised version. We cannot, of course, promise publication at that time.

If you decide to revise the manuscript for further consideration at PLOS Genetics, please aim to resubmit within the next 60 days, unless it will take extra time to address the concerns of the reviewers, in which case we would appreciate an expected resubmission date by email to plosgenetics@plos.org.

[LINK]

We are sorry that we cannot be more positive about your manuscript at this stage. Please do not hesitate to contact us if you have any concerns or questions.

Yours sincerely,

Carlos Alvarez

Guest Editor

PLOS Genetics

David Kwiatkowski

Section Editor: Cancer Genetics

PLOS Genetics

The reviewers found the work interesting but detailed major concerns with the analysis and interpretation. They also noted the high-level framing of the work could be improved greatly. For example, they suggest there is a tendency to promote the primate model as superior, and to do so without acknowledging all limitations (e.g., primate ethics and cost). We would consider a revised manuscript that satisfied those concerns.

- Carlos E. Alvarez, Guest Editor

Reviewer's Responses to Questions

**Comments to the Authors:**

Reviewer #1: This is a highly interesting description of a rhesus macaque model where some developed MLH1 deficient colorectal cancers either due to sporadic MLH1 hypermethylation or due to germline MLH1 mutations conferring a rhesus macaque version of Lynch syndrome. The authors have demonstrated the molecular basis of the 41 cases of colorectal cancer and have developed an MSI panel more specific for the rhesus macaque model. In trying to describe the rhesus macaque MLH1 Lynch syndrome animal model, I believe there are some additional details that would help clarify and characterize the authors argument that this would be the preferred animal model of study.

Are there any general details about the colony and what is the overall prevalence of colorectal cancer? Is the prevalence similar to that of ~6% of colorectal cancer in the general population?

Could the authors provide detailed pedigrees of the macaque models where germline MLH1 pathogenic variants were discovered? Were there clusters of affected animals that are related to each other over several generations? Does the transmission suggest an autosomal dominant transmission of susceptibility?

Were there other any other cancers found in the germline MLH1 macaque carriers like those found in human Lynch syndrome families? For example, were there any other animals with Lynch associated cancers (endometrial, ovarian, urothelial)?

The authors should also provide a figure of the pathologic IHC stained sections that demonstrate loss of MLH1 as well as the staining of MSH2, MSH6, and PMS2. What would also be interesting to check if high tumor infiltrating lymphocytes (TIL) counts (seen in human Lynch syndrome CRC tumors) were found in the germline MLH1 deficient tumors?

Could the authors discuss the feasibility of creating a macaque model of MSH2, MSH6, PMS, and EPCAM germline mutations?

Reviewer #2: The authors report comprehensive molecular characterization of colorectal tumors associated with a germline mutation in the MLH1 gene in Rhesus macaque. The study reveals a number of similarities between these non-human tumors and human colorectal tumors. The work is interesting. However, what types of preclinical and clinical research can be better conducted with the Rhesus model, compared to directly studying human CRC? What about the cost?

Furthermore, a number of technical issues need to be addressed

1. “CMS classification categorized rhesus CRC samples mainly as CMS2”: why they are mainly CMS2? Should the Rhesus tumors of MSI-H be classified as CMS1, like in humans?

2. Mutation burden: MSI-H colorectal tumors in human are characterized with high mutation burden; is the same observed in rhesus colorectal tumors?

3. The integration between DNA methylation and gene expression should be conducted? E.g., will promoter methylation lead to decrease in gene expression in what genes?

4. Figure 3A: PC1+PC2 is only around 25%, too small for a meaningful interpretation. The same is true for Figure 4A. 3D PCA or TSNE may be used.

5. Figure 3B: no p-values indicating how stable the clusters are.

6. More thorough comparison of each finding with that of human CRC is needed.

Reviewer #3: The authors describe the mismatch repair (MMR) status of spontaneous colorectal carcinoma (CRC) in 41 animals from a closed Rhesus macaque colony at MDACC. The results are intriguing and have the potential to add considerable value to the extant information of MMR in the pathogenesis of CRC.

The data are mostly clear, with exceptions noted below. However, critical aspects of the experiments and analysis are omitted, and the enthusiasm for the manuscript in its current form is diminished because the authors vastly over-interpret the significance and potential of what are admittedly important data, but in a model system with many more limitations than the authors acknowledge.

Specific issues of concern are enumerated below:

1. Even though the authors provide background into the colony and the presence of a group with the spontaneous MLH1 (c.1029C<g, p.tyr343ter="">

2. It seems likely that these samples were collected over a time period (which must be stated) and that the technology used to obtain the data for analysis changed over time. For example, the reviewers refer to sequencing in two different instruments. Yet, the methods do not describe batch effects and the steps taken to achieve batch correction. This is essential to interpret the methylation and the RNA sequencing data. Depending on which samples were sequenced in which instrument at which time, batch effects could be driving much of the separation along principal components and for the differentially expressed genes (DEGs). This could explain, for example, the gene expression data where MSS/MSI-L samples are indistinguishable from MSI-H samples based on hierarchical clustering using the top DEGs (Fig 4C). The methods should include a section describing the time course of sample acquisition, processing, and sequencing, as well as batch effects and methods for batch correction. Visual representation of the batch effects and the correction should be included as supplementary materials.

3. The presentation of the animals that have LS (mutant MLH1) and those that inactivate MLH1 sporadically by methylation is quite confusing. Specifically, there were 8 animals with LS. Despite the authors suggestion from Fig. 3A that these tumors segregate as a group based on their methylation status, the clustering in Fig 3C actually shows that the 4 LS animals that were analyzed (presumably the samples from the other 4 animals were not of sufficient quality or there was a process of selection that should be described) do not cluster as a group. Two samples cluster with MSI-H samples and the other two cluster with a heterogeneous group of normal and MSS/MSI-L tumors as a sub-branch of the group that includes all the normal samples. There is a clear branch that includes all the MSI-H samples and two LS samples. So, based on this, Fig 3C is a non sequitur, since the authors analyze methylation differences for all the tumors together vs. normal, when the unsupervised data already distinguish a difference between MSI-H + 2 LS vs all the other samples. The fact that all tumors have hypermethylation of TOP1, PCGF3, FAM76B, and others, and hypomethylation of ALKBH5, GAS8, MME, and many others may simply be a consequence of proliferation or the generation of a TME and is not deeply informative for the major thrust of the paper, which is describing the impact of MMR and the potential utility of the model.

4.The same applies to Fig 4, where the LS animals (how many??) are simply grouped with MSI-H and not identified separately (panel A and panel C), but they are used to compare against the TCGA data from human colon and rectal carcinomas. This seems to be inappropriate data selection. Further on Figure 4, the authors neglect to describe the data in panel C in the manuscript, although there is a reference to Fig 4D, which is not provided. Perhaps that is a typo? Although, the authors describe a defined LS cohort based on gene expression, and this is definitely not apparent (or identified by legend or color code) in Fig 4. Simply, the authors description/interpretation of the data (text) is inconsistent with the data shown in the figure (Fig 4).

5. The point above (#4) illustrates the confusion about the segregation of animals with LS and those animals with sporadic MSI-H. It seems that the tumors in the animals with LS diverge in their biology (Fig 2F and Fig 3B), perhaps less so than the sporadic MSI-H tumors, but the authors split and lump these together in the analysis and in the discussion. As a specific example, in Fig 2F the authors do not segregate the LS animals from the MSI-H animals in the description of "sporadic," as the addition of 31 MSI-H (referred to as sporadic MSI) + 6 MSI-L + 4 MSS = 41, and which means the LS animals (which would not, or should not be considered as "sporadic," but rather should be considered as "familial" or "heritable," are lumped into MSI-H (sporadic) or MSI-L categories.

6. Suppl Table 3, in particular, shows that one of the LS animals that was used to characterize the subtype of CRC was MSI-L (RM 17, acknowledged by the authors) and two were MSI-H (RM09 and RM31). RM11 also has an MSI-L phenotype (Fig 2). It seems that the same animals were not used across the analyses shown. For example, RM09 is not included in Fig. 3. It is unclear which animals are included in the analysis of Fig. 5. All of this makes Fig. 1 disingenuous, as the impression is that all animals were used for all of the analysis, except for technical exclusions.

7. The authors acknowledge that RNAseq is not the best method to identify mutations, although the point out it is "adequate." This is true when seeking to identify mutations in specific genes (rather than globally). But to make the data interpretable, the authors should include tumor content in the sample, presumed enrichment from macro-dissection, and the number of transcripts from variant alleles vs the number of transcripts from all alleles (VA/VA+WTA). Together, this information will help estimate the VAF in the tumor and provide assurance that the presumed mutations are not sequencing artifacts. Another relatively simple method to confirm the RNAseq data would be to select a random (statistically validated) sample set where mutations were identified and perform Sanger sequencing of DNA from tumor and normal for the specific region identified.

8. The data in Fig 2 needs backup. The authors should include Suppl material documenting the morphological phenotype(s) of the CRCs in the animals, as well as examples from each IHC reaction to document the interpretation of "absent" and "present." In particular, for LS animals where the assumption is that MLH1 is inactivated by a second hit in the tumor, is there expression of MLH1 and PSM2 in adjacent normal tissue (based on IHC)?

9. The data do not support the author's conclusions in lines 317 to 321. The leap of faith from the data to these conclusions is far too great.

10. The last point is that the authors advance this model recognizing some limitations (of the study), unnecessarily attacking other models (no model is "better" than any other. Each model is imperfect, and all have their own strengths). In criticizing mouse models, for example, the authors fail to acknowledge the exciting advantages and the potential of transposon-driven models. In proposing the Rhesus model for prevention, assessment of TME dynamics (which would require serial invasive interventions), development of therapies, etc., they fail to address the important and considerable limitations of practical animal numbers that would be required for such studies, the time (what is the incidence of the condition in the population and in addition having to support animals in the colony for two decades before tumors develop), the paucity of reagents for NHPs, the costs, and the ethical implications of proposing this grand vision in an era where many people would emphasize reducing animals, and especially primates, used for research. The size of the colony that would be required alone creates a sense of implausibility and makes it difficult to support the rest of the data, which have a place and an impact in helping us understand the role of mutations that affect MMR in the development and progression of CRC.</g,>

**Have all data underlying the figures and results presented in the manuscript been provided?**

Reviewer #1: Yes

Reviewer #2: None

Reviewer #3: **No: **Additional data and methods required include examples of the morphology and IHC of the tumors, a denominator for the population (size of the colony) and timeline of sample acquisition, details on experimental design for which samples were included in which experiment (and why), and batch effects for sequencing data, including methods for batch correction.

PLOS authors have the option to publish the peer review history of their article (what does this mean?). If published, this will include your full peer review and any attached files.

Reviewer #1: No

Reviewer #2: No

Reviewer #3: **Yes: **Jaime Modiano

---

## [Decision Letter · Decision Letter 1]

23 Mar 2022

Dear Dr Vilar,

We are pleased to inform you that your manuscript entitled "Comparative Molecular Genomic Analyses of a Spontaneous Rhesus Macaque Model of Mismatch Repair-Deficient Colorectal Cancer" has been editorially accepted for publication in PLOS Genetics. Congratulations!

Yours sincerely,

Carlos Alvarez

Guest Editor

PLOS Genetics

David Kwiatkowski

Section Editor: Cancer Genetics

PLOS Genetics

Comments from the Guest Editor:

I appreciate Reviewer 2's comment. I read the original criticism, and the authors' reply/revised text (I think there was a page/line numbering error in the authors' reply). My impression is that this criticism was generally addressed in the reply and revised text: clean ms. lines 334-352; tracking changes ms. lines 377-402. The authors also mentioned the following caveat in their study limitations "this study lacks pertinent information regarding the timeline of carcinogenesis..., which restricts our understanding of pre-cancer biology, and the timing of tumor development and subsequent evolution."

Dear authors: If Reviewer 2's suggestion could still improve the paper, please let us know and we'll check/approve that change.

-- Carlos Alvarez, Guest Editor

Comments from the reviewers (if applicable):

Reviewer's Responses to Questions

**Comments to the Authors:**

Reviewer #1: The authors incorporated many of my comments and suggestions to improve the manuscript.

Reviewer #2: The authors have sufficiently addressed all of my questions, except no. 1. The tumor progression stage may be a reason for the CMS2 classification. However, along with the lower TMB, I suspect these tumors may have some fundamental differences from spontaneous human CRCs. This should be more discussion on this.

Reviewer #3: The authors have addressed the reviewers's comments thoroughly and satisfactorily.

**Have all data underlying the figures and results presented in the manuscript been provided?**

Reviewer #1: Yes

Reviewer #2: Yes

Reviewer #3: Yes

PLOS authors have the option to publish the peer review history of their article (what does this mean?). If published, this will include your full peer review and any attached files.

Reviewer #1: No

Reviewer #2: No

Reviewer #3: **Yes: **Jaime Modiano

**Data Deposition**

http://datadryad.org/submit?journalID=pgenetics&manu=PGENETICS-D-21-01125R1

**Press Queries**

---

## [Editor Report · Acceptance letter]

13 Apr 2022

PGENETICS-D-21-01125R1 

Comparative Molecular Genomic Analyses of a Spontaneous Rhesus Macaque Model of Mismatch Repair-Deficient Colorectal Cancer 

Dear Dr Vilar, 

We are pleased to inform you that your manuscript entitled "Comparative Molecular Genomic Analyses of a Spontaneous Rhesus Macaque Model of Mismatch Repair-Deficient Colorectal Cancer" has been formally accepted for publication in PLOS Genetics! Your manuscript is now with our production department and you will be notified of the publication date in due course.

With kind regards,

Agnes Pap

PLOS Genetics

On behalf of:
